# Associations between Neighborhood Disadvantage and Dog Walking among Participants in the Dog Aging Project

**DOI:** 10.3390/ijerph191811179

**Published:** 2022-09-06

**Authors:** Devin Collins, Hannah Lee, Matthew D. Dunbar, Kyle Crowder

**Affiliations:** 1Department of Sociology, University of Washington, Seattle, WA 98195, USA; 2Center for Studies in Demography and Ecology, Seattle, WA 98195, USA

**Keywords:** neighborhood disadvantage, dog walking, fear of crime, built environments

## Abstract

Although neighborhood socioeconomic disadvantage is negatively related to overall physical activity, prior studies reveal a complex relationship between disadvantage and particular walking behaviors. While disadvantage is associated with reduced recreational walking through a hypothesized “fear-of-crime” mechanism, the built environment in disadvantaged neighborhoods may encourage utilitarian walking. To date, no study has assessed how disadvantage relates to dog walking, a distinct walking behavior that is neither strictly recreational nor utilitarian but represents a key mechanism through which pet ownership may affect human health. We employ a large (*n* = 19,732) dataset from the Dog Aging Project to understand how neighborhood disadvantage is associated with dog walking when controlling for individual-, household-, and environmental-level factors. We find that dog owners in more disadvantaged neighborhoods report less on-leash walking activity compared to owners in advantaged neighborhoods and discuss the possibility of a fear-of-crime mechanism underlying this association. These findings improve our understanding of the relationship between neighborhood disadvantage and physical function and highlight the need for public health interventions that encourage dog ownership to consider neighborhood disadvantage.

## 1. Introduction

A large body of literature finds that, relative to residents of advantaged neighborhoods, those living in disadvantaged neighborhoods experience higher rates of disease [1,2] and obesity [3], and report lower self-rated health [4,5]. Neighborhood disadvantage is also negatively related to physical function [6] and overall physical activity [7].

However, empirical studies also reveal a complex relationship between neighborhood disadvantage and particular walking behaviors. Namely, studies consistently find that neighborhood disadvantage is negatively associated with recreational walking but positively associated with so-called “utilitarian” walking (i.e., walking for transportation) controlling for individual-level characteristics that affect physical activity [5,7,8,9,10,11]. In a large-scale study investigating this distinction, Turrell et al. [11] found that high levels of utilitarian walking among residents of disadvantaged neighborhoods is partly attributable to low motor vehicle access and partly attributable to built environment characteristics more conducive to walking that are disproportionately present in disadvantaged neighborhoods, including street connectivity, density, and mixed land use.

At the same time, scholars have found that residents of disadvantaged neighborhoods frequently perceive their neighborhood to be less safe from crime, and that these perceptions in turn deter recreational walking [5,9]. Concerns about safety and crime victimization have been shown to be a particularly strong deterrent for recreational walking amongst older adults [12,13,14]. Advantaged neighborhoods, on the other hand, may be more likely to have aesthetic features that promote recreational walking [7,15], even if their built environments are less physically “walkable”. Leslie et al. [8], for instance, found that residents of high-socioeconomic status neighborhoods perceive their environment as more attractive, safe, and orderly, which in turn encourages greater outdoor activity, park use, and recreational walking. In sum, past studies find that disadvantaged neighborhoods are positively associated with utilitarian walking via built environment features and relatively low car ownership, while advantaged neighborhoods are positively associated with recreational walking via perceived neighborhood safety and attractiveness. In addition, they suggest that neighborhood disadvantage may reduce recreational walking by increasing fear of crime [5,16], especially for older adults [12,13]. 

While these studies offer interesting insights into neighborhood disadvantage and two types of walking behavior, no study to date has explored the relationship between neighborhood disadvantage and dog walking—a form of walking neither strictly utilitarian nor recreational. Understanding this link is important, because dog ownership has been identified as a potential low-cost public health initiative to encourage physical activity and walking frequency [17]. In 2013, the American Heart Association published a scientific statement recommending dog ownership as a strategy for promoting cardiovascular health [18]. 

Although dog ownership is positively associated with physical activity [19,20,21] and better health outcomes [22,23], there is also evidence of considerable variation in dog walking behaviors amongst owners. Past studies find that up to a third of dog owners do not walk their dogs at all, and over half walk their dogs more infrequently than is recommended for both dog and human health [24,25,26,27,28,29,30]. 

To understand what drives this variation, several studies have considered the role of neighborhood characteristics in dog walking behavior. Objective features of the built environment, including mixed land use [25], street connectivity [27,29,31], sidewalk quality [29], park access [31,32,33], and density [25,29], have all been identified as positive and significant predictors of dog walking frequency. Coleman et al. [25], for instance, find that dog owners who frequently walk their dogs are more likely to reside in neighborhoods with more “walkable” built environments compared to owners who either never walk their dogs or walk their dogs infrequently. The authors speculate that dog owners in low-walkable neighborhoods may be more likely to live in single family homes with large yards that provide space for activity and/or dog urination and defecation without needing to be walked. Subsequent studies have confirmed a negative association between dog walking frequency and home/yard size [17,26,34].

Thus, although a host of studies have explored environmental correlates of dog walking, none to our knowledge have focused on the role of neighborhood disadvantage. Given that disadvantaged neighborhoods have features that potentially both encourage walking (i.e., via higher density and built environment walkability) and discourage walking (i.e., via perceived disorder and safety concerns), it is not obvious what role neighborhood disadvantage may play in shaping dog walking. A clearer understanding of this role can help guide public health interventions seeking to promote dog ownership as a means of encouraging physical activity. 

To address these gaps, we examine the association between neighborhood disadvantage and on-leash dog walking using a unique source of data featuring a large nationwide sample of dog owners in the United States. To untangle the complex relationship between neighborhood disadvantage and various walking behaviors, we control for a number of variables at the individual-, household-, and environment-levels. Finally, we explore how the relationship between neighborhood disadvantage and dog walking varies by owner age to indirectly assess the fear-of-crime mechanism that has been observed in studies of recreational walking.

## 2. Materials and Methods

### 2.1. Sample Design

To address our research questions, we use data from the Dog Aging Project (DAP), a large-scale longitudinal study that collects survey data from participating dog owners living in the United States. The primary aim of DAP is to understand the environmental, genetic, and lifestyle determinants of healthy aging in companion dogs. Because dogs share many of the same exposures and disease risks as people, findings from the DAP also carry implications for human aging [35]. The Dog Aging Project is an open data project. These data are available to the general public at dogagingproject.org/open_data_access.

Dog owners nominate their dog for inclusion in the DAP through an online portal and are invited to complete a series of modules that together comprise the Health and Life Experience Survey (HLES). The HLES contains over two hundred survey items related to their dog’s environment, behavior, lifestyle, diet, and health. Respondents who complete all modules are invited to participate in a follow-up survey each year for the remainder of the dog’s life. The current investigation includes data collected from 27,541 respondents during the first year of the DAP, from December 2019 to December 2020 [36]. Owners of dogs not characterized as primarily companion dogs (i.e., service or working dogs) were excluded from this study (*n* = 1218). 

To supplement HLES information with objective environmental metrics, owner-reported primary residential addresses are geocoded at the rooftop or street address level. Returned residential coordinates are then used to merge HLES data with various data that characterize aspects of the respondents’ environment (described in more detail below). Respondents (*n* = 275) who did not provide geo-codable addresses were excluded from the current analysis. Respondents with missing values on any neighborhood disadvantage or key individual level characteristics were excluded. Our final sample includes 19,732 respondents. 

### 2.2. Focal Variables

#### 2.2.1. Weekly On-Leash Walking

Our focal dependent variable is an estimate of weekly on-leash dog walking frequency and duration, constructed from three separate HLES questions (Table 1). The first question is a multiple-choice prompt reading “Over the past year, if your dog leaves the house with you, they are…” and respondents are shown four options: “On-leash”, “Off-leash”, “Both”, or, “My dog doesn’t take walks.” Respondents who select either “On-leash” or “Both” are asked to estimate the average frequency that their dog is active on a leash, which is the second question that makes up our dependent variable. Six options are presented on a continuum ranging from “Less than once a month”; “Less than once a week”; “Once or twice a week”; “3–6 times a week”; “Once a day”; and “More than once a day.” Finally, respondents are asked to provide an estimate in hours and minutes for the average duration of a typical on-leash activity, the third question from which our dependent variable is constructed. 

Using these three HLES questions, we generated a continuous estimate of weekly on-leash walking duration by (1) transforming the average frequency of on-leash activity into a weekly frequency integer (i.e., transforming “once a day” to “7”), and (2) multiplying that integer by the reported duration of each walking bout (Figure 1). As we are interested in human walking behaviors in this study, we focus exclusively on on-leash dog walking, which suggests the dog owner is walking with the dog. Respondents who indicated either that their dog never takes walks or walks exclusively off-leash were coded as having zero weekly on-leash walking minutes. Our constructed on-leash walking variable produced a small number of unrealistically high weekly estimates. As a corrective, we removed respondents with levels of on-leash walking that exceeded nine hours a day on average (*n* = 33). Several sensitivity checks were performed and the inclusion or exclusion of these extreme values do not influence the interpretation of our findings. 

#### 2.2.2. Neighborhood Disadvantage

Following established convention, we employed census tract boundaries as a rough geographic approximation for the residential neighborhood. Our index of neighborhood socioeconomic disadvantage is a composite measure created by averaging the z-scores of the following five American Community Survey (ACS) variables: (1) percentage of population falling below 1.25% of the poverty line; (2) percentage of working-age (16–64) population that is unemployed in the labor force or not in labor force; (3) percentage of children living in female-led households with no adult male present; (4) percentage of population 25 years or older with less than a Bachelor’s degree as their highest level of educational attainment; and (5) percentage of households earning under $100,000 in the last 12 months. These variables were selected based on our review of prior scholarship on neighborhood effects that finds that these characteristics, which are frequently associated with one another, together form reliable and valid approximations of neighborhood disadvantage [37,38,39]. All census data used in this study were derived from the 2014–2018 5-year ACS. 

### 2.3. Covariates

To estimate the independent role of neighborhood disadvantage on dog walking, we controlled for an array of dog-, owner-, household-, and contextual-level characteristics (Table 2). These characters were selected based on previous findings in the relevant literature or otherwise plausibly associated with dog walking. 

#### 2.3.1. Dog-, Owner-, and Household-Level Characteristics 

Dog-, owner-, and household-level characteristics were sourced from HLES. Dog age and dog size were included, as both are established as strongly associated with physical function and activity among dogs [17,40,41,42].

At the level of individual owners, we controlled for age and income, both initially reported in ranges and transformed to continuous (the middle value of each range) for ease of interpretation. For example, respondents selecting the age range “35–44” were re-coded as 39.5 years old. In addition, we controlled for owner education (a binary variable indicating attainment of Bachelor’s degree or higher) and race/ethnicity (a binary variable indicating non-Hispanic white). Our selection of owner-level controls was based on prior research indicating that these factors influence physical dog walking specifically and physical function generally [25,26,28,40,43,44]. 

Finally, we controlled for several household-level characteristics that have been shown to influence dog walking [17,25,26,34,40,45]. These include homeownership status (whether the respondent rents, owns, or other); dwelling type (operationalized as a binary variable indicating single family home vs. other); yard size (in square footage, zero if dog has no yard access); and household composition (a binary variable indicating whether the respondent lives in a single-person household, as well as a numerical indicator of how many children are present in the home). 

#### 2.3.2. Contextual-Level Characteristics 

We controlled for several contextual-level characteristics plausibly associated with both neighborhood disadvantage and on-leash dog walking. Contextual measures come from several sources. First, two relevant environment-level survey items are asked in the HLES: (1) whether there are parks in the respondents’ neighborhood (Yes/No); and (2) whether there are sidewalks available in the respondents’ neighborhood. For this latter item, three options are available: “Yes on all streets”, “Yes, on some streets”, and “No, on no streets”.

The remainder of the environmental-level covariates came from secondary data sources that objectively characterize the neighborhood. First, we included the Walk Score for each geocoded coordinate. Walk Score is a popular proprietary algorithm that provides a walkability score ranging from 0–100 based on area density, street layout, and distance to various amenities [46]. Walk Score has been validated by numerous researchers as an indicator of neighborhood walkability [47]. 

At the tract level, we controlled for population density (from the 2014–2018 5-year ACS, persons per square mile) and air pollution. Air pollutant concentration data came from public-use estimates developed by the Center for Air, Climate and Energy Solutions (CACES) using v1 empirical models as described by Kim et al. [48]. CACES provides ten-year estimates of various air pollutants at the tract-level, with the most recent data from 2015. Estimates for the aerosol pm2.5 are used in the current study. 

Lastly, we controlled for temperature and precipitation at the county level, derived from NOAA’s Climate Divisional Database (nClimDiv). We included total county rainfall (in inches) from the previous year (2019), as well as a measure of temperateness constructed by taking the difference of the highest and lowest monthly average air temperature recorded in the previous year. Descriptive statistics for the final sample are provided in Table 2.

### 2.4. Analytic Strategy

Treating our dependent variable (weekly on-leash walking frequency and duration) as a continuous measure, we employed ordinary least squares (OLS) regression modeling to estimate the relationship between neighborhood disadvantage and on-leash walking. Although distribution of our dependent variable is skewed, OLS produces reliable estimates given a large sample size such as ours [49]. To address heteroskedasticity, we used heteroskedasticity-consistent standard error estimates (HC3) across our models [50].

First, we fitted a univariate regression model including the disadvantage index as the only explanatory variable. In subsequent models we sequentially added terms for dog-, owner-, household-, and contextual-level characteristics. Collinearity amongst explanatory variables was measured using variance inflation factors, which suggested no significant multicollinearity. 

We also tested for an interaction effect between owner age and neighborhood disadvantage. As Ross and Mirowsky [5] note, the relationship between neighborhood disadvantage and walking is likely explained through a fear-of-crime mechanism. Disadvantaged neighborhoods are frequently characterized by greater disorder, higher objective crime rates, and worse subjective perceptions of safety [5,51]. In addition, fear of crime has been shown to be a particularly strong predictor of walking behaviors for certain demographic groups, such as women and older adults [9,13,14,52]. In the current analysis, we were unable to account for owner gender, perceived neighborhood disorder, and objective crime rate/fear of crime because the HLES does not include a question for owner’s gender and we did not have access to data on perceptions of neighborhood disorder and crime rates. However, we were able to assess the extent to which there is an interaction between owner age and neighborhood disadvantage. If a fear of crime mechanism is behind an observed association between neighborhood disadvantage and reduced walking frequency, we hypothesize that the negative association would be stronger for older owners than for younger owners.

## 3. Results

Of the 19,732 participants with complete data, 18,648 (94.5%) reported that their dog walks on-leash, 631 respondents (3.2%) reported that their dog walks off-leash only, and 453 (2.3%) respondents reported that their dog does not take walks. The average reported duration of individual on-leash walks is 54.3 min, and average weekly on-leash walking time is an estimated 411.9 min per week (6.9 h). 

The results of all regression model fits are presented in Table 3. In Model 1, weekly on-leash walking was regressed on neighborhood disadvantage alone. Neighborhood disadvantage has a strong negative association with on-leash walking; one standard deviation higher neighborhood disadvantage corresponds with 79 fewer weekly on-leash walking minutes on average.

In Model 2, all dog-, owner-, and household-level covariates were included to account for additional factors that may influence dog walking and its association with neighborhood disadvantage. Controlling for these characteristics reduced the size of the neighborhood disadvantage coefficient. One standard deviation higher neighborhood disadvantage corresponds with an average approximately one hour less weekly on-leash walking, controlling for dog-, owner-, and household-level characteristics. 

In Model 3, all contextual-level confounders were included to further isolate the association with neighborhood disadvantage. The coefficient for neighborhood is only marginally attenuated compared to Model 2; when built environment characteristics, air quality, temperature, and precipitation factors are controlled, one standard deviation higher neighborhood disadvantage corresponds with 55 fewer weekly on-leash walking minutes. This suggests that neighborhood disadvantage is significantly and independently associated with on-leash walking, controlling for other environmental features like Walk Score, population density, and sidewalk/park availability. 

Finally, Model 4 tests for an interaction between disadvantage and owner age. The interaction coefficient is negative, suggesting that, all else equal, higher levels of neighborhood disadvantage suppress on-leash walking to a greater extent for older owners. This interaction is visualized in Figure 2, which displays predicted levels of weekly on-leash walking by neighborhood disadvantage and owner age categories based on Model 4. Consistent with other analyses using these data [41], owner age tends to be positively related to dog activity; on-leash walking duration is higher for older owners than younger owners at all levels of neighborhood disadvantage (Figure 2). As neighborhood disadvantage index increases, the predicted weekly on-leash walking duration declines more rapidly for older owners than for younger owners, though this interaction is only statistically significant at the 0.1 level (*p* = 0.068).

## 4. Discussion

To our knowledge, this is the first study to explore the relationship between neighborhood disadvantage and dog walking. While neighborhood effects researchers have previously explored the role of neighborhood disadvantage on utilitarian and recreational walking, they have not explored dog walking specifically. Conversely, while a handful of studies assessing determinants of dog walking have included environmental covariates, neighborhood disadvantage has heretofore not been examined. 

Using cross-sectional data from a large-scale study of dog owners in the U.S., we find that residents of relatively disadvantaged neighborhoods report substantially less on-leash walking, controlling for dog-, owner-, and household-level characteristics. In addition, after controlling for potential environmental confounders, we continue to observe an independent association between neighborhood disadvantage and reported on-leash walking. Specifically, each standard deviation increase in the disadvantage index is associated with approximately 55 fewer walking minutes per week in the full model. In other words, owners in relatively more disadvantaged neighborhoods are walking their dogs on average almost one hour less per week than those in relatively advantaged neighborhoods. If valid, this finding has potentially important implications for an owner’s health given the American Heart Association’s recommendation of 150 min of moderate exercise per week. 

The observed association between neighborhood disadvantage and dog walking behavior resembles patterns more similar to recreational walking than to utilitarian walking (i.e., walking for transport). Residents of disadvantaged neighborhoods may perceive their neighborhoods to be less safe from crime, which discourages recreational walking in these contexts, particularly for older individuals [9,13,14]. In the current study, we find that the inverse association between neighborhood disadvantage and dog walking is modified by owner age, i.e., the association is stronger for older owners than younger owners. One interpretation of this observation is that a fear-of-crime mechanism underlies the observed association between neighborhood disadvantage and dog walking. That is, the effect of neighborhood disadvantage on walking might be particularly pronounced among older residents because it is among these individuals that fear of crime is most salient [12]. However, the weak statistical significance of this interaction provides only preliminary and tentative evidence for this theory. Future studies should test this possibility by directly measuring residential perceptions of crime, safety, and disorder and assessing their connection to dog walking behaviors.

## 5. Conclusions

This study has several limitations. First and most importantly, our data are cross-sectional, imposing limitations on causal inference. Second, our focus on on-leash walking may introduce bias towards a more urban/less rural environment. Third, Several owner-level characteristics that have been shown to influence dog walking were excluded from this analysis due to data limitations, including gender [31,40,52]; owner’s intrinsic motivation and obligation [28,32,42,43]; and health status [25]. Future studies exploring the link between residential environment and dog walking should include these measures, particularly gender, as previous research has found perceived neighborhood safety to be an important predictor of dog walking among women but not men [52]. 

Additionally, though we include an annual measure of temperature extremity, our study does not account for more granular ecological trends such as heatwaves or snowstorms that may play a role in dog-walking patterns. Likewise, some neighborhood characteristics such as quality of walkable destinations and perceived neighborhood aesthetics [27,45] are likely to have an important influence on dog walking frequency, but could not be included in the present analysis due to data constraints. For instance, Sugiyama et al. [15] found that the availability of parks with dog-supportive amenities, such as litter bags and trash bins, may matter more than the mere presence or absence of neighborhood parks. Future studies should consider whether access to such features attenuates the relationship between neighborhood disadvantage and on-leash walking described in the current study.

Because DAP respondents make up a nonrandom sample of American dog owners who voluntarily elect to participate in the HLES, our data capture a unique subset of dog owners who are more affluent, more well-educated, and whiter than the general US population. As noted above, some HLES respondents also reported very high weekly on-leash walking, and although the inclusion/exclusion of these outliers does not change the interpretation of our findings, they raise some potential measurement validity concerns. Finally, there is substantial overlap between the data collection period and the onset of COVID-19, which may have had significant impacts on on-leash dog walking activity amongst owners. Longitudinal studies incorporating future waves of DAP data will be able to speak to any COVID-related impacts in the current study, assess the robustness of the neighborhood disadvantage-walking relationship, and more rigorously determine whether neighborhood disadvantage causally precedes dog walking behavior. 

Despite these limitations, the substantial association between neighborhood disadvantage and dog walking has implications for both dog and human physical function. More research is needed to understand the mechanisms driving this association and to guide appropriate policy responses. If residents in disadvantaged neighborhoods are found to be discouraged from walking their dogs due to safety concerns and/or a lack of dog-friendly infrastructure, the development of dog parks in these neighborhoods could be considered. Aside from promoting dog walking, dog parks have been shown to foster community interaction, trust, and social capital [53,54,55]. At the same time however, dog parks can also be a signal or even cause of neighborhood gentrification, potentially promoting social or racial exclusivity [56]. Thus, any planning initiatives should include the participation of residents (i.e., through community-led development) to ensure interventions benefit all community members and reflect their concerns [55,57]. 

Taken together, our study highlights that public health interventions seeking to encourage physical well-being through dog walking should be attendant to larger political and economic processes that drive and sustain neighborhood inequality. Meaningfully addressing structural disinvestment and environmental stressors may be just as or more effective for promoting physical health for residents of disadvantaged neighborhoods as dog ownership.

## Figures and Tables

**Figure 1 ijerph-19-11179-f001:**
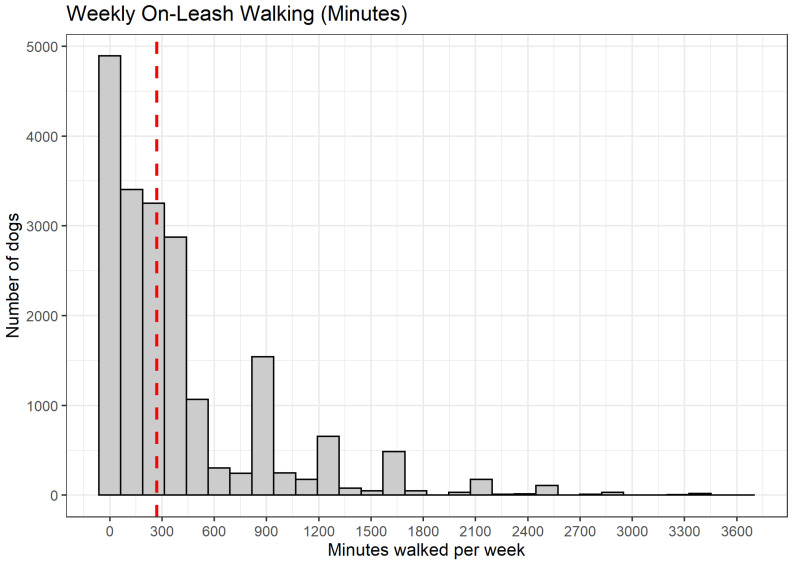
Histogram of Weekly On-Leash Walking Minutes. Vertical red line corresponds to median weekly walking minutes.

**Figure 2 ijerph-19-11179-f002:**
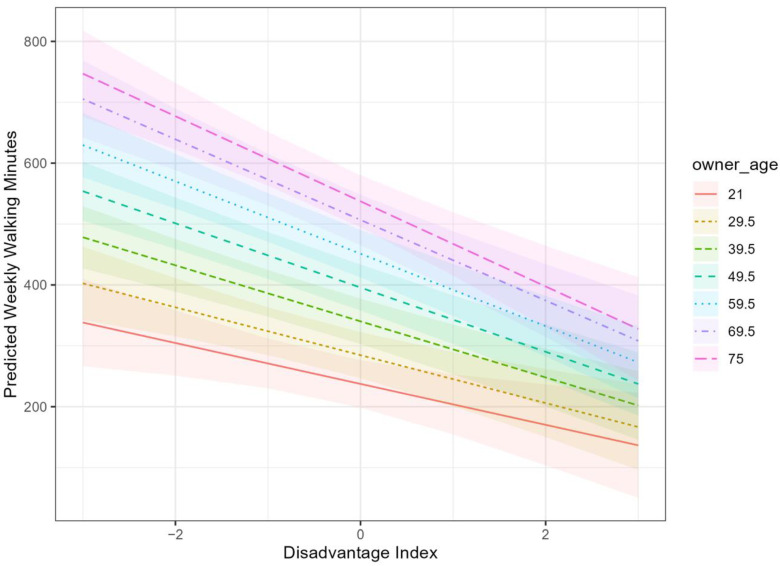
Predicted Weekly Walking Minutes by Age and Neighborhood Disadvantage from Model 4.

**Table 1 ijerph-19-11179-t001:** Distribution of Dog Aging Project Participants by Components of Summary “Dog Walking” Variable (*n* = 19,732). Dog Aging Project, 2019–2020.

	Descriptive Statistics *n* (%)
*Question 1: Walking Type ^a^*	
My dog doesn’t take walks	453 (2%)
On-leash	11,141 (57%)
Off-leash	631 (3%)
Both	7507 (38%)
*Question 2: On-Leash Walking Frequency ^b^*	
Less than once a month	1237 (6%)
Less than once a week	1388 (7%)
Once or twice a week	2275 (12%)
3–6 times a week	3172 (16%)
Once a day	3533 (18%)
More than once a day	7043 (36%)
Not Applicable (Dog does not take on-leash walks)	1084 (6%)
*Question 3: Average On-Leash Walk Duration (Minutes) ^c^*	
Mean (SD)	54.3 (37.5)
Median [Min, Max]	40 [0, 530]
Not Applicable (Dog does not take on-leash walks)	1084 (6%)
*Dependent Variable: Weekly On-Leash Walking (Minutes)*	
Mean (SD)	411.9 (492.9)
Median [Min, Max]	270 [0, 3640]

^a^ Question text: “Over the past year, if your dog leaves the house with you, they are…” ^b^ Question text: “Over the past year, what is the average frequency that your dog is active on a lead/leash?”. The qualitative options for this question were transformed, in order, into the following weekly frequency estimates: 0.167, 0.5, 1.5, 4.5, 7, 14. ^c^ Question text: “On average, how long does that activity last?”.

**Table 2 ijerph-19-11179-t002:** Descriptive Statistics of Neighborhood Disadvantage Index and Control Variables (*n* = 19,732). Dog Aging Project, 2019–2020.

Characteristic	*n* (%), Mean (SD), or Median (Min, Max)
*Disadvantage Index*	
Mean (SD)	−0.512 (0.641)
Median [Min, Max]	−0.560 [−2.20, 2.74]
*Dog Age*	
Mean (SD)	7.23 (4.18)
Median [Min, Max]	7.00 [0, 24.8]
*Dog Weight (lbs)*	
Mean (SD)	50.2 (29.0)
Median [Min, Max]	50.0 [0.170, 230]
*Owner Age*	
Mean (SD)	52.4 (14.4)
Median [Min, Max]	49.5 [21.0, 75.0]
*Income ($)*	
Mean (SD)	112,000 (51,000)
Median [Min, Max]	110,000 [10,000, 180,000]
*BA or Higher Education*	
Yes	15,805 (80%)
No	3927 (20%)
*Non-Hispanic White*	
Yes	17,757 (90%)
No	1975 (10%)
*Homeownership Status*	
Rent	2702 (14%)
Own	16,726 (85%)
Other	304 (2%)
*Single Family Home*	
Yes	16,451 (83%)
No	3281 (17%)
Yard Size (ft^2^)	
Mean (SD)	24,600 (40,000)
Median [Min, Max]	3000 [0, 131,000]
*Single Person Household*	
Yes	3882 (20%)
No	15,850 (80%)
*Number of children*	
Mean (SD)	0.259 (0.662)
Median [Min, Max]	0 [0, 8.00]
*Neighborhood Parks*	
Yes	16,450 (83%)
No	3282 (17%)
*Sidewalk Availability*	
No Streets	6801 (35%)
Some Streets	4189 (21%)
All Streets	8742 (44%)
*Walk Score*	
Mean (SD)	30.4 (29.0)
Median [Min, Max]	22.0 [0, 100]
*Population Density (persons per square mile)*	
Mean (SD)	4280 (9680)
Median [Min, Max]	2200 [0.430, 198,000]
*pm2.5*	
Mean (SD)	7.51 (1.69)
Median [Min, Max]	7.54 [2.33, 15.5]
*Rainfall (in)*	
Mean (SD)	43.4 (14.4)
Median [Min, Max]	46.7 [6.04, 90.6]
*Temperature Range (°F)*	
Mean (SD)	61.7 (10.2)
Median [Min, Max]	64.0 [32.8, 95.7]

**Table 3 ijerph-19-11179-t003:** Estimated associations between neighborhood disadvantage and dog walking. Dog Aging Project, 2019–2020.

	Model 1	Model 2	Model 3	Model 4
	Estimate, (95% CI)	Estimate, (95% CI)	Estimate, (95% CI)	Estimate, (95% CI)
Disadvantage Index	**−78.89, (−89.67, −68.05)**	**−59.36, (−70.56, −48.17)**	**−54.53, (−65.79, −43.27)**	−19.49, (−56.56, 17.58)
Dog Age		**−15.54, (−17.13, −13.94)**	**−15.72, (−17.31, −14.14)**	**−15.72, (−17.31, −14.13)**
Dog Weight (lbs)		**0.43, (0.21, 0.65)**	**0.45, (0.22, 0.67)**	**0.45, (0.23, 0.67)**
Owner Age		**5.72, (5.20, 6.24)**	**5.86, (5.34, 6.38)**	**5.55, (4.95, 6.15)**
Annual Household Income		**0.00, (0.00, 0.00)**	**0.00, (0.00, 0.00)**	**0.00, (0.00, 0.00)**
Bachelor’s degree or Higher		**41.36, (24.20, 58.52)**	**34.38, (17.32, 51.44)**	**34.29, (17.23, 51.35)**
Non-Hispanic White		−8.79, (−30.54, 12.95)	−0.50, (−22.28, 21.29)	−0.92, (−22.70, 20.86)
Homeownership Status (Own)		**−50.17, (−74.32, −26.02)**	−34.39, (−58.41, −10.37)	−33.94, (−57.96, −9.92)
Homeownership Status (Other)		−5.31, (−64.50, 53.88)	18.37, (−40.69, 77.42)	19.70, (−39.38, 78.78)
Single Family Home		**−169.97, (−193.42, −146.52)**	**−133.83, (−157.85, −109.80)**	**−134.60, (−158.63, −110.56)**
Yard Size (ft^2^)		**−0.00, (−0.00, −0.00)**	**−0.00, (−0.00, 0.00)**	**−0.00, (−0.00, 0.00)**
Single Person Household		27.36, (7.85, 46.87)	23.96, (4.60, 43.33)	24.37, (5.00, 43.74)
Number of Children		**−44.49, (−53.27, −35.71)**	**−43.62, (−52.32, −34.92)**	**−43.14, (−51.85, −34.43)**
Neighborhood Parks			**83.57, (66.87, 100.28)**	**83.32, (66.61, 100.03)**
Sidewalk Availability (Some Streets)			**35.17, (14.55, 55.78)**	**34.71, (14.10, 55.33)**
Sidewalk Availability (All Streets)			**34.58, (15.27, 53.89)**	**34.49, (15.18, 53.81)**
Walk Score			**1.11, (0.77, 1.44)**	**1.11, (0.77, 1.44)**
Population Density			0.00, (0.00, 0.00)	0.00, (0.00, 0.00)
pm2.5			−5.88, (−10.16, −1.60)	−6.01, (−10.29, −1.73)
Rainfall (in)			0.77, (0.28, 1.27)	0.78, (0.29, 1.28)
Temperature Range			**−1.23, (−1.95, −0.51)**	**−1.24, (−1.96, −0.51)**
Disadvantage Index × Owner Age				−0.67, (−1.40, 0.05)
Constant	**371.49, (363.00, 379.97)**	**323.07, (283.83, 362.31)**	**227.21, (154.58, 299.84)**	**244.85, (170.66, 319.03)**
Akaike Information Criterion (AIC)	300,481.2	298,993.1	298,733.3	298,731.9
Observations	19,732	19,732	19,732	19,732

Note: Bolded values indicate statistical significance at *p* ≤ 0.001.

## Data Availability

The Dog Aging Project is an open data project. These data are available to the general public at dogagingproject.org/open_data_access.

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
