# Peer review of "Associations between Neighborhood Disadvantage and Dog Walking among Participants in the Dog Aging Project"

_ijerph, 2022, doi:10.3390/ijerph191811179_

Round 1

Reviewer 1 Report

Interesting study. Impressive that the authors could analyze such a large data set with so many variables. Although temperature range noted, I was wondering if areas with large amounts of snowfall/or icy sidewalks in the winter and conversely heatwaves in the summer was taken into account. Other limitations were noted and conclusions warranted.

1. What is the main question addressed by the research? The main question is what is posed in the title; i.e. the associations btw neighborhood disadvantage and dog walking 2. Do you consider the topic original or relevant in the field, and if
so, why? This is relevant to public health as recommendations to increase walking via dog walking need to take into account neighborhood disadvantage.
3. What does it add to the subject area compared with other published
material? To my knowledge this is the first study to examine this relationship (and it is an important relationship to dissect)
4. What specific improvements could the authors consider regarding the
methodology? Methodology is appropriate and limitations were noted. The only further suggestion I can offer would be to take into account seasonal changes in walkability such as at the extremes of temperature (heatwaves in summer and ice/snow/freezing temps in the winter). Not really sure how that would be accomplished and may only be able to cite this as another limitation.
5. Are the conclusions consistent with the evidence and arguments
presented and do they address the main question posed? Yes
6. Are the references appropriate? Quite thorough
7. Please include any additional comments on the tables and figures. Data was quite complex and the tables and figures did a good job of distilling the information into readable formats.

Author Response

Response to reviewer 1:

Thank you for your review and suggestions. We agree that seasonal weather patterns likely play a role in walking that is not captured by our “temperature range” variable. To address this limitation, we have added the following language on line 333:

  • “… though we include an annual measure of temperature extremity, our study does not account for more granular ecological trends such as heatwaves or snowstorms that may play a role in dog-walking patterns.”

Reviewer 2 Report

Thank you for the opportunity review this paper. Overall, the authors have crafted a well written manuscript with current references and unique approaches. I only have minor concerns regarding the manuscript.

First, the histogram appears to show some outliers in the data that likely should be removed from the dataset. For example, on participant appears to be walking their dog for over 8 hours a day. While the authors address skewness of the variables, it is hard to believe that the other answers by individuals who are outliers like the one I mention are reliable. I thank the authors for addressing the skew, but I am having difficulty believing that any dog owners are walking their dogs on the upwards of 4 hours a day each week. Regression is a powerful tool, but data cleaning is still essential for ensuring trustworthiness in its results.

Second, the article offers novel and insightful results; however, greater discussion is needed in two areas:

1) The data is based on the Dog Aging Project Consortium. More information about the consortium, its aims, and in general the implications on aging and older adulthood in disadvantaged neighborhoods could be added to the introduction and conclusion.

2) The authors could share more regarding implications for policy makers and practitioners engaging in community development work and leadership in disadvantaged neighborhoods. Some example references regarding community development and dogs are below to explore, but may not need to be included. They are just examples:

Booth, A. L. (2017). Dog eat dog world: public consultation and planning on contested landscapes, a case study of dog parks and municipal government. Community Development Journal52(2), 337-353.

Meijer, M. (2019). Community-led and government-fed:: Comparing informal planning practices in depopulating regions across Europe. Journal of Rural and Community Development14(4).

Krumholz, N., Keating, W. D., Star, P. D., & Chupp, M. C. (2006). The long-term impact of CDCs on urban neighborhoods: Case studies of Cleveland's Broadway-Slavic Village and Tremont neighborhoods. Community Development37(4), 33-52.

Author Response

Thank you for your review and insightful comments. We have made several changes to address your feedback. First, you raise concerns about some of the very high levels of on-leash walking reported by some survey respondents. We are similarly puzzled by these outliers, and further investigation is needed to understand these respondents’ dog-walking behaviors and to ensure measurement validity. We have addressed this limitation on line 347:

  • “As noted above, some HLES respondents also reported very high weekly on-leash walking, and although the inclusion/exclusion of these outliers does not change the interpretation of our findings, they raise some potential measurement validity concerns.”

To your second point, we have included some additional language about the Dog Aging Project on line 100:

  • “The primary aim of DAP is to understand the environmental, genetic, and lifestyle determinants of healthy aging in companion dogs. Because dogs share many of the same exposures and disease risks as people, findings from the DAP also carry implications for human aging [35].”

The linked reference (35) by Creevy et al. (2022) provides further detail about the aims and  scope of the Dog Aging Project.

Finally, we thank you for your suggested references and your recommendation to include more specific policy implications. We have addressed this on line 358 by adding the following language:

  • “More research is needed to understand the mechanisms driving this association and to guide appropriate policy responses. If residents in disadvantaged neighborhoods are found to be discouraged from walking their dogs due to safety concerns and/or a lack of dog-friendly infrastructure, the development of dog parks in these neighborhoods could be considered. Aside from promoting dog walking, dog parks have been shown to foster community interaction, trust, and social capital [53-55]. At the same time however, dog parks can also be a signal or even cause of neighborhood gentrification, potentially promoting social or racial exclusivity [56]. Thus, any planning initiatives should include the participation of residents (i.e., through community-led development) to ensure interventions benefit all community members and reflect their concerns [55, 57].”